# A pruned and parameter-efficient Xception framework for skin cancer classification

Şafak Kılıç 1,2☯*, Yahya Doğan 3☯

1 School of Computer Science, CHART Laboratory, University of Nottingham, Nottingham, United Kingdom, 2 Faculty of Engineering, Architecture and Design, Department of Software Engineering, Kayseri University, Kayseri, Turkey, 3 Department of Computer Engineering, Siirt University, Siirt, Turkey

☯ These authors contributed equally to this work.
* safakkilic@kayseri.edu.tr

## Abstract

Skin cancer is one of the most prevalent and potentially lethal diseases worldwide, with early detection being critical for patient survival. This study presents a novel framework that leverages transfer learning, pruning, SMOTE, data augmentation, and the advanced Avg-TopK pooling method to improve the accuracy and efficiency of skin cancer classification using dermoscopic images. The HAM10000 dataset was used to evaluate the performance of various transfer learning models, with Xception as the top performer. A layer-based pruning strategy was proposed to optimize the model and reduce its complexity. SMOTE and data augmentation were applied to address the class imbalance within the dataset, significantly improving the model's generalization across all skin lesion classes. The utilization of the Avg-TopK pooling technique further enhanced model accuracy by preserving crucial image features during the downsampling process. The proposed approach achieved an overall accuracy of 91.52%, surpassing several state-of-the-art models. Following pruning, the model's parameter count was reduced by approximately 35%, from 20.9 million to 13.5 million, improving efficiency and performance. This framework demonstrates the effectiveness of combining model pruning, oversampling, and advanced pooling methods to build robust and efficient skin cancer classification systems suitable for clinical applications.

## 1. Introduction

Skin cancer is a rapidly growing concern in the medical field due to its increasing incidence worldwide, driven by factors such as environmental changes, radiation exposure, lifestyle habits, and genetic predispositions [1,2]. Among all cancer types, skin cancer holds a dominant position, with millions of new cases diagnosed annually. The complexity of skin lesions, which vary greatly in appearance and often resemble benign conditions, makes early detection challenging for even the most experienced dermatologists [3,4]. This difficulty highlights the need for more advanced diagnostic

**Data availability statement:** Dataset was obtained from the publicly accessible Kaggle repository (https://www.kaggle.com/datasets/kmader/skin-cancer-mnist-ham10000). All authors accessed the data under the same conditions as any other researcher, and no special access privileges were used.

**Funding:** The author(s) received no specific funding for this work.

**Competing interests:** The authors have declared that no competing interests exist.

tools, especially as some types, like melanoma, are highly aggressive and life-threatening if not detected early [5].

The introduction of dermoscopy as a non-invasive imaging technique has significantly enhanced the accuracy of skin cancer diagnosis. However, its effectiveness is still dependent on the expertise of dermatologists, which varies widely [2,6]. As skin cancer cases continue to rise, automated methods using deep learning, particularly Convolutional Neural Networks (CNNs), have gained traction for their ability to accurately classify skin lesions based on image data [1,3]. CNNs have revolutionized the field by reducing the need for handcrafted features, allowing faster and more reliable diagnoses in real-time settings [7,8].

Transfer learning models have been particularly effective in medical image classification tasks, leveraging pre-trained networks such as Xception, DenseNet, and ResNet to reduce training time and computational resources while maintaining high accuracy [9]. However, these models often involve many parameters, which can be computationally expensive and difficult to deploy in real-time or resource-constrained environments. To address this, pruning techniques are increasingly being explored to reduce the number of parameters in these models without significantly compromising their performance [10,11]. Applying pruning to the highest-performing transfer learning model enables the creation of a more efficient model that retains high accuracy while reducing computational load.

One of the significant challenges encountered in skin cancer classification is the issue of class imbalance, where certain lesion types are underrepresented in the dataset. This imbalance can lead to biased predictions, with the model favoring majority classes. To address this problem, techniques such as SMOTE (Synthetic Minority Over-sampling Technique) and ADASYN (Adaptive Synthetic Sampling) are employed in various problem domains to generate synthetic examples for minority classes, thereby balancing the dataset and enabling the model to produce more accurate results across all classes [12–14]. Additionally, data augmentation techniques such as rotation, scaling, and color adjustments are applied to increase the diversity of the dataset, thus preventing overfitting. The HAM10000 dataset used in this study faces class imbalance issues, and comprehensive balancing and data processing techniques must be applied to obtain a more robust and accurate model.

Recent advancements in 2025 have further expanded the scope of deep learning in oncological applications, ranging from skin lesion classification to breast cancer detection, emphasizing the need for both high accuracy and computational efficiency. For instance, Aruk et al. [15] introduced a novel hybrid approach combining ConvNeXt blocks with Vision Transformers (ViT), which captures both local textural patterns and global long-range dependencies, achieving 94.30% accuracy on the HAM10000 dataset. In a comprehensive comparative study, Aruk et al. [16] benchmarked 15 CNNs against 15 ViT models, demonstrating that while Swin Transformer-based ViTs yield superior accuracy (92.12%), they incur higher computational costs compared to CNNs. Beyond dermatology, similar architectural optimizations are critical in other domains; Alswilem and Pacal [17] highlighted the trade-off between diagnostic performance and efficiency in breast cancer ultrasound analysis, identifying

RexNet-200 as a pragmatic, high-efficiency model. Additionally, Cakmak and Pacal [18] demonstrated the efficacy of InceptionV3 in breast ultrasound classification, achieving 96.67% accuracy, which further underscores the importance of selecting appropriate deep learning architectures for specific medical imaging tasks.

In addition to addressing the class imbalance, enhancing model architecture through optimized pooling strategies can further boost performance. Traditional pooling layers, such as max pooling or average pooling, are commonly used in CNNs to reduce the dimensionality of feature maps. However, replacing these layers with advanced techniques like Avg-TopK Pooling can improve feature selection by retaining the most informative data points, leading to more accurate predictions [19]. This approach enhances the model's accuracy and reduces the risk of losing important features during the downsampling process.

This study investigates the performance of multiple transfer learning models on a large skin cancer dataset. The top-performing model is then pruned to create a lightweight version, maintaining high performance while reducing the number of parameters. In addition, the class imbalance problem in the dataset is addressed through the use of SMOTE and data augmentation techniques. Traditional pooling layers are replaced with Avg-TopK pooling to evaluate its impact on overall model performance. These optimizations aim to develop an efficient and high-performing model for skin cancer classification, suitable for real-time diagnostic applications.

This study makes the following key contributions:

- In this study, a novel parameter-efficient framework for skin cancer classification was developed using pruning, SMOTE, data augmentation, and advanced Avg-TopK pooling.

- A sparsity-based pruning strategy was proposed to reduce the complexity of the best-performing model. This pruning method decreased the number of parameters by approximately 35% (from 20.9 million to 13.5 million), resulting in a more parameter-efficient variant of Xception while still maintaining high classification performance.

- Experiments were carried out on a well-known dataset, i.e., HAM10000, demonstrating that the proposed method improves classification accuracy by addressing class imbalance through SMOTE and enhancing the model's generalization with data augmentation techniques.

- The integration of the advanced Avg-TopK pooling technique further boosted model performance by preserving essential features during the downsampling process, resulting in improved accuracy and robustness in skin cancer classification.

- The performances of a pruned Xception model were compared to standard models through both quantitative metrics and qualitative evaluations, demonstrating that the pruned model retains high accuracy while significantly reducing the number of parameters, making it more efficient and practical for clinical applications.

The structure of this article is organized as follows: Section 2 reviews the existing literature on skin cancer classification techniques. Section 3 describes the proposed approach, including the use of transfer learning, pruning, SMOTE, data augmentation, and Avg-TopK pooling. Section 4 presents the experimental setup and analyzes the results. Finally, the article concludes with a discussion of the findings and potential directions for future research.

## 2. Related works

In medical image analysis, particularly in the classification of skin diseases, a wide range of techniques have been applied to enhance the accuracy and reliability of diagnostic systems [20,21]. Each technique offers distinct advantages and faces specific challenges depending on the complexity of the dataset and the nature of the task. From traditional statistical methods to advanced neural networks, these approaches are designed to tackle the diverse characteristics of skin lesions, such as texture, color, and shape. Several widely used techniques, as summarized in Table 1, are discussed, highlighting their key functions and limitations in the context of skin disease classification.

**Table 1. Summary of related works on skin lesion classification, including classical methods, hybrid approaches, and deep learning models.**

| Studies | Technique | Function | Limitations |
|---|---|---|---|
| [22–24] | Morphological Operations | Morphological operations, such as dilation and erosion, are used to enhance key structural features of an image. These operations help detect boundaries and regions indicative of abnormalities by manipulating the image's shape based on structuring elements. | Determining optimal threshold values is challenging. Subtle lesion changes may be overlooked, and results may vary depending on lesion shape, texture, and size, limiting performance in complex skin disease tasks. |
| [26–28] | Gray Level Co-occurrence Matrix (GLCM) | GLCM analyzes texture by calculating how often pixel pairs with specific intensities occur at given spatial relationships, enabling extraction of robust textural features. | Computationally demanding; extracted features may not be invariant to rotation, scaling, or texture variations, reducing robustness under dynamic conditions. |
| [29–31] | Bayesian Classifier, Decision Tree, KNN, SVM, ANN | Bayesian classifier handles discrete/continuous data efficiently. Decision Trees capture nonlinear rules. KNN uses similarity-based classification. SVM maximizes class margins. ANN models complex nonlinear relationships. | Bayesian classifier struggles with dependent predictors. Decision Trees may overfit. KNN loses performance with noisy/large datasets. SVM requires careful tuning. ANN may suffer from overfitting, spatial information loss, and vanishing/exploding gradients. |
| [32–34] | Genetic Algorithm | Optimization technique inspired by natural evolution that uses crossover/mutation to explore solution space efficiently. | May not converge to global optimum; can be computationally expensive, especially for high-dimensional or complex problems. |
| [36–38] | Convolutional Neural Networks | CNNs automatically extract hierarchical features from images through convolutional layers, capturing intricate visual patterns effectively. | Struggles with variations in object scale/position. Requires large computational resources and long training times. Sensitive to spatial shifts and transformations. |
| [39–41] | Ensemble Models | Combines strengths of multiple models, improving accuracy and robustness by capturing diverse data patterns. Useful for complex tasks. | Higher risk of overfitting. Low interpretability. High computational cost. May struggle with unexpected data discrepancies. |

Morphological operations have been widely used as preprocessing techniques to enhance structural features such as lesion boundaries and abnormal regions. By applying operations like dilation and erosion, these methods aim to emphasize diagnostically relevant shapes. However, their effectiveness strongly depends on threshold selection and structuring elements, making them less reliable for complex skin lesion variations in shape, texture, and size [22–25].

Gray Level Co-occurrence Matrix (GLCM) is a statistical texture analysis method that extracts spatial intensity relationships between pixel pairs. While effective for capturing textural patterns, GLCM-based approaches are computationally expensive and lack robustness to rotation, scale, and texture variations, limiting their generalization performance [26–28].

Traditional machine learning algorithms, including Bayesian classifiers, Decision Trees, KNN, SVM, and ANN, have been extensively applied to skin disease classification. These methods leverage handcrafted features related to texture, color, and shape. Despite their effectiveness in structured scenarios, they often suffer from overfitting, sensitivity to feature selection, and limited scalability when dealing with complex and high-dimensional image data [29–31].

Genetic Algorithms (GA) have also found applications in skin disease classification tasks due to their ability to explore large and complex solution spaces efficiently. In skin classification, GA is utilized to optimize feature selection, model parameters, and other classification criteria. GA helps identify near-optimal solutions for distinguishing between different skin conditions by simulating natural selection, crossover, and mutation processes. However, despite its strengths, GA does not always guarantee convergence to the global optimum, especially in high-dimensional spaces, and its computational cost can be substantial, making it less ideal for real-time medical applications [32–34].

CNNs have become one of the most widely used techniques in skin disease classification due to their ability to automatically learn and extract relevant features from images. In particular, CNNs excel in handling the complexities of image data, making them highly suitable for tasks such as skin lesion classification. A common approach within this field involves leveraging transfer learning, where pre-trained CNN models such as ResNet, Inception, or VGG are fine-tuned on specific

datasets, significantly reducing training time while achieving high accuracy. Despite their effectiveness, CNNs come with certain limitations [35]. They sometimes struggle to interpret the scale and size of objects within images, which is crucial in distinguishing subtle differences in skin lesions. Moreover, CNNs demand substantial computational resources and long training times to achieve reliable performance. Additionally, maintaining spatial invariance ensuring that small changes in object position or orientation do not impact the model's predictions remains a challenge, potentially affecting the robustness of the classification results [36–38].

Ensemble learning approaches improve skin lesion classification performance by combining multiple classifiers or deep models to enhance robustness and accuracy. These methods benefit from complementary feature representations but introduce increased computational complexity, reduced interpretability, and a higher risk of overfitting when training data variability is high [39–41].

While CNN-based approaches have established strong baselines, recent literature from 2024 and 2025 highlights a shift towards Vision Transformers (ViT) and hybrid architectures to capture global dependencies. For instance, Lian et al. introduced a shifted windowing vision transformer that achieved 93.60% accuracy on the HAM10000 dataset, demonstrating the efficacy of attention mechanisms over pure CNNs [42]. Similarly, Agarwal and Mahto proposed a hybrid CNN-Transformer model utilizing Convolutional Kolmogorov-Arnold Networks (CKAN) for feature fusion, which reached an accuracy of 92.81% [43]. Recently, Kılıç (2025) proposed FocusGate-Net, a dual-attention guided MLP–convolution hybrid segmentation network that integrates shifted token MLP blocks with CBAM and attention gates, achieving strong performance and cross-dataset generalization on ISIC2018, PH2, and Kvasir-SEG datasets while maintaining high computational efficiency [44]. Furthermore, Yang et al. developed a multi-scale attention booster within an ensemble framework, achieving a remarkable 95.05% accuracy by focusing on discriminative lesion regions [45]. These studies collectively indicate that integrating global context modeling with local feature extraction represents the current state-of-the-art.

## 3. Materials and methods

### 3.1. Dataset

This study used the publicly available skin cancer HAM10000 dataset [46] to classify skin cancer. This dataset consists of 10,015 dermatoscopic images, including actinic keratoses (akiec), basal cell carcinoma (bcc), benign keratosis-like lesions (bkl), dermatofibroma (df), melanoma (mel), melanocytic nevi (nv), and vascular lesions (vasc). However, as shown in Fig 1, there is a noticeable class imbalance in the dataset. Most images belong to the nv class, accounting for 6,705 images, making up a significant portion of the dataset. In contrast, minority classes such as vasc and df have only 142 and 115 images, respectively.

This imbalance between classes may pose challenges in model training, as the model is likely to become biased towards the majority class, leading to better performance for this class while struggling to classify the minority classes correctly. To address this imbalance, techniques such as data augmentation, class weighting, or oversampling of minority classes can be employed to achieve a more balanced model training process.

The images in the dataset are originally in RGB format with a resolution of $600 \times 450$ pixels. In this study, the images were resized to $128 \times 128$ pixels to ensure compatibility with the model's input size and reduce computational complexity. While the HAM10000 dataset provides a valuable resource for automatically classifying skin lesions, it is crucial to consider the class imbalance during model development. Fig 2 presents example images from each class in the dataset. These images illustrate the diversity of skin lesions and highlight the visual differences between the various classes.

### 3.2. Ethical considerations

This study did not involve any new data collection from human participants. All experiments were conducted using the publicly available HAM10000 dataset [46], which consists of fully de-identified and anonymized dermoscopic images.

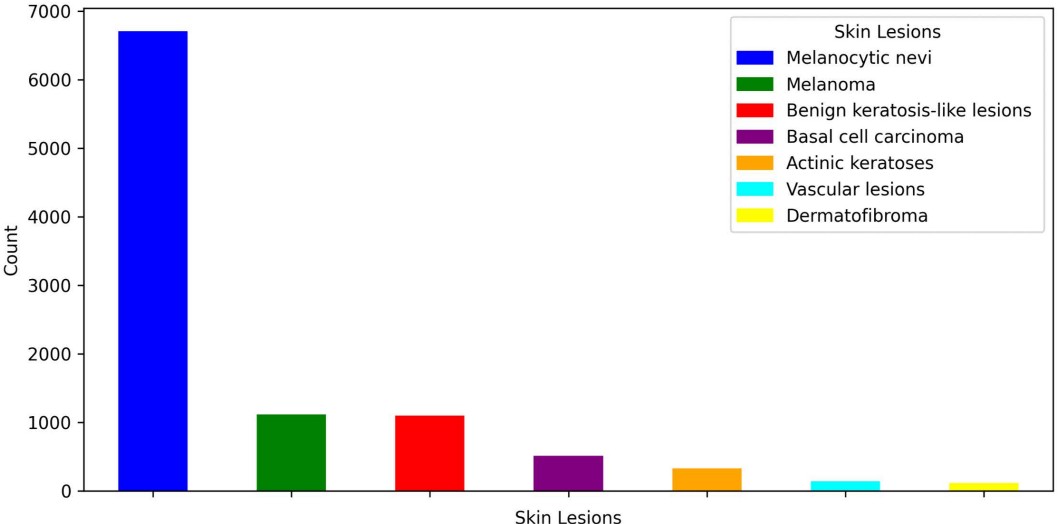

**Fig 1. Distribution of skin lesion classes in the HAM10000 dataset.** The dataset exhibits a significant class imbalance, with most images belonging to the melanocytic nevi class, while minority classes such as dermatofibroma and vascular lesions are underrepresented.

According to the original dataset publication, image acquisition and data collection were performed in compliance with applicable ethical standards and were approved by the relevant institutional review boards. As this study exclusively used secondary, anonymized data, no additional ethical approval or informed consent was required.

### 3.3. Transfer learning models

Transfer learning enables the use of pre-trained deep neural networks—originally trained on large-scale datasets—for new tasks with limited data. By leveraging previously learned representations, training time is reduced and generalization is improved. Each model utilized in this study incorporates architectural innovations that enhance feature extraction, computational efficiency, and convergence behavior. This subsection summarizes the key contributions and mathematical foundations of the models employed.

**DenseNet201** introduces dense connectivity, where each layer receives the concatenated outputs of all preceding layers. This formulation facilitates feature reuse, strengthens gradient flow, and reduces redundancy across layers. Formally, the output of layer $l$ is expressed as:

$$x_l = H_l([x_0, x_1, ..., x_{l-1}]),$$ (1)

where $H_l$ denotes the composite transformation at layer $l$. Dense connectivity mitigates vanishing gradients and enhances representational efficiency.

**VGG16** employs a deep architecture with small $3 \times 3$ convolution filters stacked sequentially to capture fine-grained spatial features. A single convolution operation is defined as:

$$y_{ij} = \sum_{m=-1}^{1} \sum_{n=-1}^{1} w_{mn} x_{(i+m)(j+n)} + b,$$ (2)

where $y_{ij}$ denotes the output activation, and $w_{mn}$ the filter weights. Despite its simplicity, VGG16 provides a strong baseline for feature extraction.

| vasc | mel | nv | df | bkl | bcc | akiec |
|------|-----|----|----|-----|-----|-------|

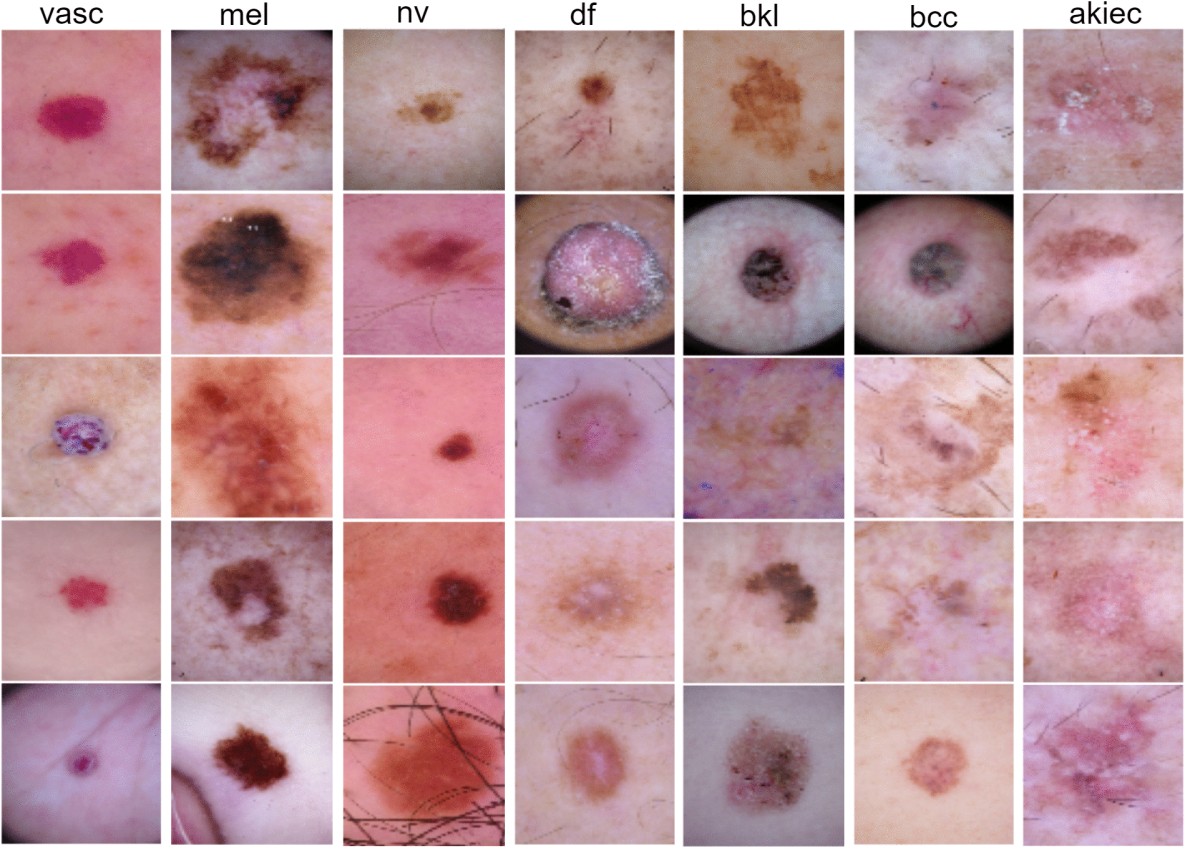

**Fig 2. Example images for each class from the HAM10000 dataset.** The images illustrate the visual diversity of skin lesion categories and highlight inter-class variability across different lesion types.

**ResNet50** resolves the vanishing gradient issue by introducing residual learning. Residual blocks add shortcut connections that allow gradients to propagate directly through the network:

$$y_l = F(x_l, W_l) + x_l, \tag{3}$$

where $F$ denotes the residual mapping. This design stabilizes training in deeper networks and improves convergence.

**MobileNetV2** is built upon depthwise separable convolutions, decomposing a standard convolution into depthwise and pointwise operations. The depthwise step is:

$$y_{ij} = \sum_k w^k x_{ijk}, \tag{4}$$

followed by a pointwise convolution:

$$z_{ij} = \sum_m w'_{ij} y_{ijm}. \tag{5}$$

This decomposition dramatically reduces computational cost, making MobileNetV2 suitable for resource-limited environments.

**EfficientNetB3** applies a compound scaling strategy that uniformly scales network depth, width, and resolution using predefined coefficients:

$$d \to \alpha d, \quad r \to \beta r, \quad w \to \gamma w, \tag{6}$$

where $\alpha$, $\beta$, and $\gamma$ are tuned scaling constants. This balanced scaling yields improved performance with fewer parameters.

**Xception** extends the idea of depthwise separable convolutions by fully separating spatial and channel-wise correlations. The depthwise operation is:

$$y_{ij} = \sum_k w^k x_{ijk}, \tag{7}$$

followed by a pointwise projection:

$$z_{ij} = \sum_m w'_{ij} y_{ijm}. \tag{8}$$

This architecture enhances efficiency without compromising accuracy, making it highly effective for transfer learning tasks.

**InceptionV3** extracts multi-scale features by applying convolution filters of different receptive field sizes in parallel:

$$y_{1 \times 1} = W_{1 \times 1} * x, \quad y_{3 \times 3} = W_{3 \times 3} * x, \quad y_{5 \times 5} = W_{5 \times 5} * x. \tag{9}$$

This parallel structure enables the model to capture both local and global patterns, improving representational richness. In summary, the models employed in this study incorporate complementary innovations—such as dense connectivity, residual learning, separable convolutions, and multi-scale feature extraction—that collectively enhance transfer learning performance. These architectures allow efficient reuse of pre-trained features and facilitate rapid adaptation to the HAM10000 dataset.

## 3.4. The proposed pruning method

Model pruning is a widely used optimization technique to enhance the performance of deep learning models. This technique aims to optimize a model's speed and memory usage by removing unnecessary or low-importance parameters. In large models, some parameters may contribute minimally to the output. The pruning process identifies these redundant parameters and reduces the model's size without compromising performance. This approach is especially important for mobile, edge, and other environments with limited hardware resources.

In CNN models, several pruning methods are commonly used, including Weight-based Pruning, Structured Pruning, and Layer Pruning. Weight-based Pruning [47,48] selects which weights to remove based on their magnitude. Weights with smaller values generally contribute less to the model's output, so they are either set to zero or entirely removed. This method is effective in significantly reducing the number of parameters without greatly impacting the model's accuracy. It introduces sparsity into dense networks, which improves computational efficiency. Structured Pruning [49,50], unlike weight-based pruning, targets entire structures within the model, such as neurons or filters. For instance, in a convolutional layer, entire filters or channels can be removed. This approach not only reduces the model's size but also creates a hardware-friendly structure, which facilitates easier parallel computations. Structured pruning is especially useful in environments with hardware constraints, like mobile or edge devices, as it enhances inference speed and reduces memory usage. Layer Pruning [51,52] focuses on removing whole layers or blocks within the model. In architectures like ResNet, layers that do not significantly contribute to performance may be deactivated or entirely removed. This technique can dramatically decrease the model's size and complexity, particularly in deep networks where some layers become redundant as training progresses, leading to more efficient computations.

In this study, a novel pruning method for CNNs is proposed, which reduces model depth by analyzing layer-wise sparsity after training. Instead of relying on a single test image, sparsity values are now computed using a validation batch of 16 images. For each layer, activations are collected for all samples in this batch, and sparsity is defined as the proportion of zero activations averaged across the batch. This provides a more stable and representative estimate of layer importance.

Sparsity for each layer $l_i$ is computed as:

$$S(l_i) = \frac{1}{B} \sum_{b=1}^{B} \frac{\sum(x_b == 0)}{x_b.size},$$

(10)

where $B = 16$ denotes the validation batch size, and $x_b$ represents the activations for layer $l_i$ for the $b$-th image in the batch. Once the sparsity values for all layers are calculated, the layer with the highest sparsity is identified. The critical step in the pruning process involves removing all layers following the layer with the highest sparsity. The detailed pruning process is outlined in Algorithm 1. Formally, the pruned model $L_{\text{pruned}}$ consists of all layers up to and including the layer with the highest sparsity:

$$L_{\text{pruned}} = \{l_i \in L \mid i \leq \text{argmax}(S(l_i))\}$$

(11)

where $\text{argmax}(S(l_i))$ denotes the index of the layer with the highest sparsity. After the pruning operation, the modified model is retrained from scratch to adapt to its reduced architecture. The retraining process allows the pruned model to learn new representations without the removed layers, potentially leading to a more compact model that maintains its original performance levels. It should be noted that this retraining step is not a separate neural architecture search process; rather, it is an integral part of the pruning strategy that ensures the reduced architecture can effectively learn after the pruning decision guided by batch-averaged sparsity.

This pruning method differs from traditional techniques that target individual neurons or weights. By analyzing layer-wise sparsity and removing entire sections of the network, a more drastic reduction in model complexity is achieved. This approach is particularly advantageous in cases where deeper layers contribute minimally to the final output. Retraining the pruned model allows for maintaining strong performance while significantly reducing computational cost. The objective is not to produce an ultra-lightweight architecture, but to obtain a parameter-efficient reduction relative to the original Xception structure.

**Algorithm 1** Layer-wise Sparsity-Based Pruning

```
Input Trained CNN model model, validation batch X = {x₁, ..., x₁₆}
Output Pruned and retrained model pruned_model
 1:  sparsity_values ← [ ]
 2:  For each layer lᵢ in model do
 3:    batch_sparsity = 0
 4:    For each image x_b in X do
 5:      a_{i,b} ← GETACTIVATIONS (lᵢ, x_b)
 6:      batch_sparsity+ = ∑(a_{i,b} = 0) / |a_{i,b}|
 7:    end for
 8:    S(lᵢ) = batch_sparsity/|X|
 9:    APPEND (sparsity_values, S(lᵢ))
10: end for
11: k ← argmaxⱼ S(lⱼ)                    ▷ index of max-sparsity layer
12: pruned_model ← SLICELAYERS(model, 1:k)
13: RETRAIN (pruned_model)
14: return pruned_model
```

### 3.5. Handling imbalanced datasets

Many classification algorithms assume that the dataset used for training is balanced, meaning that each class has a similar number of observations. However, in real-world scenarios, datasets often exhibit class imbalance, where one class has significantly more observations than the other. In such cases, models tend to become biased toward the majority class, leading to poor performance in predicting the minority class, which is often the class of greater interest. This bias occurs because standard classification models aim to minimize the overall error rate, inherently placing more weight on the majority class due to its larger presence in the dataset. As a result, the minority class is often overlooked, leading to misclassification or under-representation in the final model predictions. This issue has drawn significant attention in recent years due to its implications in various fields such as medical diagnostics, fraud detection, and rare event prediction.

Several strategies have been developed to address class imbalance, one of the most popular being resampling methods. Resampling modifies the dataset to balance the class distributions either by oversampling the minority class or undersampling the majority class [53]. Studies have demonstrated that balancing class distributions through resampling can improve model performance in imbalanced settings [54,55].

Resampling techniques are generally divided into undersampling, oversampling, and hybrid methods. In undersampling methods, observations from the majority class are randomly removed until the classes are balanced. Random undersampling is the simplest form of this approach, where majority class instances are randomly discarded, reducing bias but potentially leading to loss of valuable information. On the other hand, oversampling methods aim to increase the representation of the minority class. Random oversampling achieves this by duplicating minority class instances and adding them to the dataset [56]. Although Random oversampling is straightforward, it can lead to overfitting as the model may become too focused on specific minority class samples, which can reduce generalization performance on unseen data.

To overcome this, [57] introduced SMOTE, a more advanced method that generates synthetic samples of the minority class. Rather than simply duplicating existing instances, SMOTE creates new samples by interpolating between existing minority class instances and their nearest neighbors. This process helps the model to generalize better by covering the feature space more effectively, reducing overfitting. In recent years, various SMOTE variants have been developed to address specific limitations and challenges of the original method [58–61]. These variants aim to further improve the performance of oversampling, particularly in complex and highly imbalanced datasets. For instance, ADASYN [58], which adapts the generation of synthetic samples based on the density of minority class instances, gives more attention to harder-to-classify regions of the data. SMOTEWB [61] incorporates a noise detection mechanism combined with a boosting procedure to mitigate the generation of synthetic samples in noisy regions. This approach determines the appropriate number of neighbors for each observation, reducing the risk of overfitting or generating unrealistic data points. The method leverages the strengths of boosting to improve class balance and overall model robustness, proving to be effective in noisy environments. These SMOTE variations provide tailored solutions for different scenarios, enhancing the robustness and effectiveness of resampling techniques in handling imbalanced datasets.

Hybrid methods combine both oversampling and undersampling approaches to leverage the strengths of each method, aiming for a balanced and representative dataset without sacrificing generalization. By adopting these resampling techniques, it becomes possible to address the class imbalance and improve the classifier's ability to perform well in both classes.

In this study, the SMOTE technique was utilized under various scenarios. The first of these strategies is the 'minority' option, which ensures that only the minority class is oversampled. Another strategy, 'not minority', focuses on oversampling all classes except the minority class, while 'not majority' allows for oversampling of all classes except the majority class. Furthermore, the model performance was evaluated by generating different numbers of samples for the minority classes, such as $2\times$, $3\times$, and $4\times$ oversampling. SMOTE can be applied to images with a resolution of $128 \times 128 \times 3$, where each image consists of three color channels (RGB). The technique generates synthetic samples in the

high-dimensional pixel space by interpolating between existing minority class samples. For an image $x_i$ represented as a vector in the $128 \times 128 \times 3$ feature space, SMOTE creates synthetic images based on the following formulation:

$$x_{new} = x_i + \lambda \times (x_{nn} - x_i) \tag{12}$$

Where $x_i$ is a vector representing the original minority class image, $x_{nn}$ is one of the $k$-nearest neighbors of $x_i$ within the minority class, also represented as a vector in the $128 \times 128 \times 3$ space. The term $\lambda$ is a random scalar between 0 and 1, ensuring that the new synthetic sample lies somewhere between $x_i$ and $x_{nn}$, effectively interpolating between the two vectors in the feature space. This interpolation is performed for each pixel in the image. Since each pixel has three values (one for each color channel: Red, Green, and Blue), the interpolation is applied independently to each channel. Thus, for each pixel $p$ in the image, the corresponding pixel in the new synthetic image is calculated as:

$$p_{new} = p_i + \lambda \times (p_{nn} - p_i) \tag{13}$$

Where $p_i$ is the pixel value of the original image and $p_{nn}$ is the corresponding pixel value in the neighbor image. This process generates a new synthetic image within the feature space defined by the minority class, creating more variation and reducing the risk of overfitting specific samples. By applying SMOTE in this high-dimensional space, the minority class is better represented, helping the model generalize more effectively during training.

## 3.6. Data augmentation

One of the most significant challenges faced by machine learning and deep learning algorithms is the lack of sufficient training data. This issue often results in overfitting, where the model performs exceptionally well on training data but fails to generalize to unseen data. Overfitting occurs when the model effectively memorizes the training data instead of learning the underlying patterns, causing poor performance when it encounters new inputs. To mitigate this problem, data augmentation is widely used, especially in image-based tasks. Data augmentation artificially increases the size of the training dataset by applying various transformations to the existing data, thus helping the model to generalize better by exposing it to new variations of the same data.

In this study, several augmentation techniques were applied using the following transformations: scaling the pixel values by rescaling (1./255), rotating images by up to 10 degrees, shifting the width and height by 20%, applying shear transformations with a range of 0.2, flipping the images both horizontally and vertically, and filling missing pixels using the nearest neighbor interpolation method. These transformations introduce variability in the training data, allowing the model to learn more robust features. For instance, rotation and flipping help the model to recognize objects regardless of their orientation, while shifting and shearing introduce positional variation. This increases the overall diversity of the dataset without requiring the collection of new data from scratch, which can be particularly useful in fields like medical imaging, where acquiring large datasets is often challenging and time-consuming.

Moreover, another challenge faced is the issue of imbalanced datasets, where some classes contain significantly more images than others. As shown in Table 2, certain classes have far fewer images, and this imbalance can lead to poor model performance on the minority classes, a problem known as class misclassification. Data augmentation helps alleviate this by artificially balancing the dataset, especially for the underrepresented classes, and improving the model's ability to generalize across all classes, even the minority ones.

## 3.7. Avg-TopK pooling method

Avg-TopK pooling is a novel pooling method designed to address the limitations of traditional pooling techniques like max and average pooling. In this method, the average of the top $K$ values within a pooling window is taken, which

**Table 2. Number of training samples for each class after applying different sampling strategies.**

| Sampling Strategy | nv | mel | bkl | bcc | akiec | vasc | df |
|---|---|---|---|---|---|---|---|
| Original | 6705 | 1113 | 1099 | 514 | 327 | 142 | 115 |
| No Sampling-Train | 5331 | 908 | 872 | 420 | 272 | 114 | 95 |
| Not majority | 5331 | 5331 | 5331 | 5331 | 5331 | 5331 | 5331 |
| Minority | 5331 | 908 | 872 | 420 | 272 | 114 | 5331 |
| Not minority | 5331 | 5331 | 5331 | 5331 | 5331 | 5331 | 95 |
| Case 1 - 2x | 5331 | 1916 | 1744 | 840 | 544 | 228 | 190 |
| Case 2 - 3x | 5331 | 2724 | 2626 | 1260 | 816 | 342 | 285 |
| Case 3 - 4x | 5331 | 3832 | 3488 | 1680 | 1088 | 456 | 380 |

helps preserve significant information while allowing the model to utilize more representative features. Let $X$ be the set of input values in a pooling window of size $n \times n$. The top $K$ highest values in the pooling window are represented by $Y_1, Y_2, \ldots, Y_K$, such that:

$$Y_1 \geq Y_2 \geq \cdots \geq Y_K \qquad (14)$$

Where $Y_1$ is the maximum value, $Y_2$ is the second-highest value, and so on. The Avg-TopK pooled value ($X_{\text{Avg-TopK}}$) is calculated by averaging the top $K$ values:

$$X_{\text{Avg-TopK}} = \frac{1}{K} \sum_{i=1}^{K} Y_i \qquad (15)$$

Unlike max pooling, which selects only the maximum value, Avg-TopK pooling preserves more information by averaging the top $K$ values, thus retaining multiple important features from the input. Consider a $3 \times 3$ pooling window with the following values:

$$\begin{bmatrix} 1 & 5 & 2 \\ 3 & 8 & 6 \\ 4 & 7 & 0 \end{bmatrix}$$

If $K = 3$, the top 3 values are $8, 7, 6$. The Avg-TopK pooled value is calculated as:

$$X_{\text{Avg-TopK}} = \frac{1}{3}(8 + 7 + 6) = \frac{21}{3} = 7 \qquad (16)$$

In this example, Avg-TopK pooling retains more information compared to max pooling (which would only select $8$), and it provides a more meaningful output than average pooling (which would compute $\frac{36}{9} = 4$).

### 3.8. Training details

In this section, the details of the training process used to optimize the classification model are presented. The dataset images were resized to a resolution of $128 \times 128 \times 3$. The dataset was split into training and testing sets, following the common practice of using 80% of the data for training and 20% for testing. This split aligns with previous research utilizing similar datasets, ensuring consistency in methodology [4].

The model was compiled using the Adam optimizer with a learning rate of 0.001. The loss function chosen for this classification task was sparse categorical cross entropy, as the target labels were encoded as integers representing the different skin classes. The model's performance was tracked using the accuracy metric, which provides a clear indication of how well the model distinguishes between the skin classes.

To avoid overfitting and ensure optimal training, early stopping was implemented. Early stopping monitors the validation loss during training and halts the process if there is no improvement for 10 consecutive epochs. Additionally, a ReduceL-ROnPlateau callback was employed to reduce the learning rate by a factor of 0.1 if the validation loss plateaued for more than three epochs. This adaptive learning rate strategy ensures that the model continues to improve during later stages of training.

The training process was conducted for a maximum of 50 epochs, using a batch size of 16. During training, 20% of the training data was set aside as a validation set to monitor the model's performance and optimize hyperparameters through early stopping and learning rate adjustments.

## 4. Experiment and results

This section presents the experimental studies conducted to improve the performance of skin cancer classification. First, several widely used transfer learning models were evaluated on the original HAM10000 dataset. The models included DenseNet201, VGG16, ResNet50, MobileNetV2, EfficientNetB3, Xception, InceptionV3, and InceptionResNetV2. Their performance metrics—accuracy, precision, recall, F1-score, early stopping epoch, and parameter count—are summarized in Table 3, reported as mean ± standard deviation over five independent runs. The results show that Xception continued to outperform the other models, achieving an accuracy of 84.70 ± 0.30%, precision of 84.50 ± 0.33%, recall 446 of 84.68 ± 0.29%, and an F1-score of 84.55 ± 0.31%. Training consistently converged around the 17th epoch, demonstrating stable and fast optimization behavior. Despite having approximately 20.9 million parameters, Xception exhibited the most balanced and reliable performance across evaluation metrics.

In contrast, VGG16 remained the lowest-performing model, with an accuracy of 68.47 ± 0.55% and an F1-score of 55.76 ± 0.73%. Its relatively shallow architecture and limited representational capacity contributed to slower convergence and higher training loss. MobileNetV2 and EfficientNetB3 continued to provide strong efficiency–accuracy trade-offs, with accuracies above 83% and relatively low parameter counts, particularly in the case of MobileNetV2 (2.3M parameters). Figs 3 and 4 provide additional insights into model behavior. In Fig 3, models such as Xception and EfficientNetB3 show rapid loss reduction early in training, while VGG16 maintains higher loss values throughout. Early stopping prevented overfitting in all models, with most converging between epochs 15 and 20. Fig 4 demonstrates the fast and stable rise in accuracy for Xception and EfficientNetB3, whereas VGG16 shows slower and lower stabilization consistent with its overall performance. In summary, Xception remained the strongest baseline model across repeated runs, followed by

**Table 3. Performance scores of different models for the original dataset (mean ± standard deviation over 5 runs).**

| Models | Accuracy (%) | Precision (%) | Recall (%) | F1-Score (%) | Early Stopping | Parameters |
|---|---|---|---|---|---|---|
| DenseNet201 | 80.51 ± 0.32 | 79.85 ± 0.40 | 80.47 ± 0.35 | 80.05 ± 0.37 | 23. epoch | 18,445,383 |
| VGG16 | 68.47 ± 0.55 | 47.15 ± 0.80 | 68.52 ± 0.60 | 55.76 ± 0.73 | 15. epoch | 14,747,975 |
| ResNet50 | 78.30 ± 0.41 | 76.92 ± 0.48 | 78.28 ± 0.39 | 77.40 ± 0.44 | 19. epoch | 23,719,303 |
| MobileNetV2 | 83.05 ± 0.36 | 82.49 ± 0.42 | 83.02 ± 0.38 | 82.56 ± 0.40 | 19. epoch | 2,340,423 |
| EfficientNetB3 | 83.60 ± 0.34 | 83.05 ± 0.39 | 83.57 ± 0.36 | 83.25 ± 0.37 | 17. epoch | 10,882,358 |
| **Xception** | **84.70 ± 0.30** | **84.50 ± 0.33** | **84.68 ± 0.29** | **84.55 ± 0.31** | **17. epoch** | **20,938,543** |
| InceptionV3 | 74.61 ± 0.47 | 70.84 ± 0.52 | 74.59 ± 0.45 | 71.95 ± 0.49 | 48. epoch | 21,934,375 |
| InceptionResNetV2 | 80.10 ± 0.38 | 78.52 ± 0.43 | 80.07 ± 0.36 | 78.85 ± 0.40 | 19. epoch | 54,435,559 |

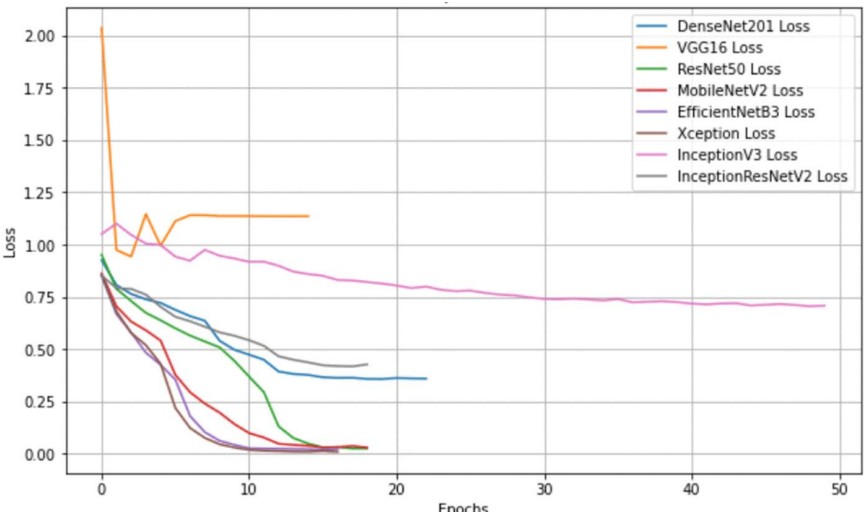

**Fig 3. Training loss curves of multiple deep learning models.** The X-axis denotes training epochs, while the Y-axis represents loss values. All models were trained with early stopping to mitigate overfitting. Most architectures exhibit a stable reduction in loss, whereas VGG16 shows slower convergence, highlighting differences in learning dynamics across models.

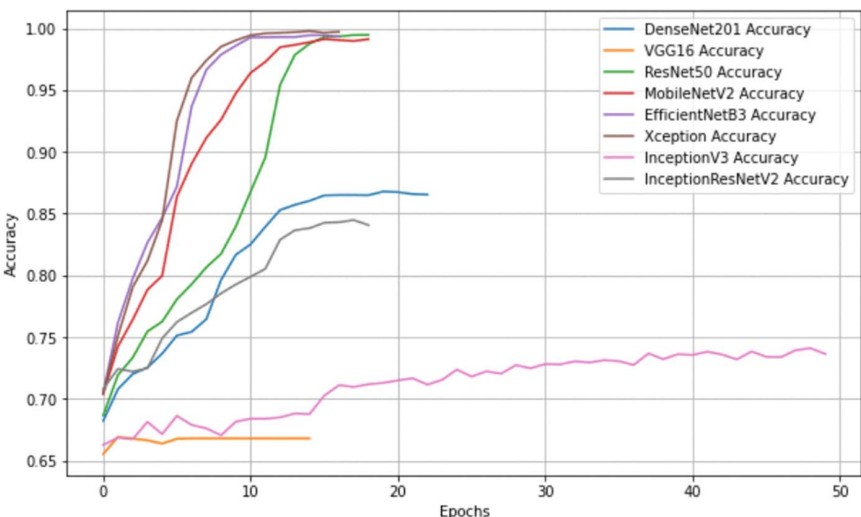

**Fig 4. Training accuracy curves of different deep learning models.** The X-axis represents training epochs, while the Y-axis indicates accuracy. Most architectures demonstrate a rapid increase in accuracy during the early epochs, with DenseNet201, Xception, and EfficientNetB3 approaching near-perfect performance. In contrast, VGG16 converges more slowly and stabilizes at a lower accuracy level, reflecting differences in training dynamics among models.

EfficientNetB3 and MobileNetV2. The updated metrics confirm the robustness and consistency of these models and support the selection of Xception as the base architecture for subsequent pruning and optimization steps.

After evaluating the performance of various transfer learning models, the Xception model was identified as the best-performing model. The next step involved making the Xception model more efficient by reducing redundant layers while preserving performance. To achieve this, the pruning method described earlier, which analyzes layer-wise

sparsity to identify layers that contribute less to the model's overall performance, was applied. Based on the sparsity values, the layers with the highest sparsity were progressively pruned, and the model was retrained after each pruning operation. During this process, eight layers with the highest sparsity values were identified, and the model was pruned starting from the layer with the highest sparsity. After pruning, the model was retrained from scratch for each configuration to allow it to adapt to the reduced architecture. In addition, the ablation pipeline follows the sequence *Pruning → SMOTE → Augmentation* to ensure methodological consistency. Pruning is performed first because it fixes the final architecture; SMOTE is applied afterward to generate synthetic samples in the correct feature space; and augmentation is applied last to enrich visual diversity for the finalized model. This ordering ensures stable, architecture-aligned, and reproducible training across all ablation stages. Table 4 presents the performance scores and the corresponding parameter reductions after pruning the model based on the proposed layer-wise sparsity. As shown in Table 4, the model maintained high-performance levels after progressively removing layers with the highest sparsity values. The best performance was achieved after pruning the model at the *block12_sepconv3_act* layer, with an accuracy of 85.33% ± 0.24, precision of 85.12% ± 0.26, recall of 85.30% ± 0.23, and an F1-Score of 84.88% ± 0.25. This pruning resulted in a model with 13.5 million parameters, a significant reduction compared to the original 20.9 million parameters in the unpruned Xception model, representing a reduction of approximately 35% in the number of parameters.

The pruning strategy revealed a significant relationship between the sparsity rates of individual layers and the model's overall performance. Specifically, layers exhibiting higher sparsity rates demonstrated a reduced contribution to the network's accuracy, thereby enabling their removal with minimal impact on performance. For instance, pruning the layer with the highest sparsity rate, *block12_sepconv3_act* (sparsity rate: 91.3%), resulted in an accuracy increase to 85.33% and a substantial reduction in the number of parameters (to 13.5 million). This finding indicates that higher sparsity rates are associated with lower layer importance, allowing for more aggressive pruning strategies without compromising performance. Conversely, pruning layers with lower sparsity rates, such as *block14_sepconv1_act* (sparsity rate: 79.6%), led to a smaller performance gain. These results confirm that focusing on layers with higher sparsity rates can produce more efficient models while preserving competitive accuracy levels. In conclusion, this method successfully reduced the Xception model's computational complexity by carefully analyzing layer-wise sparsity while maintaining competitive performance. The proposed pruning method led to a more compact model, reducing the number of parameters by up to 35%, making the model significantly more efficient for deployment in environments with limited computational resources. While the pruned model remains larger than compact architectures such as MobileNetV2 (2.3M parameters), it offers a substantial reduction compared to the original Xception (20.9M parameters). Therefore, the contribution lies in achieving parameter efficiency within the Xception family rather than competing with ultra-lightweight models.

**Table 4. Performance scores for different layers with sparsity for the original dataset (mean ± standard deviation over 5 runs).**

| Layers | Sparsity Rate (%) | Accuracy (%) | Precision (%) | Recall (%) | F1-Score (%) | Early Stopping | Parameters |
|---|---|---|---|---|---|---|---|
| block14_sepconv1_act | 79.6 | 83.55 ± 0.28 | 82.90 ± 0.32 | 83.50 ± 0.30 | 83.00 ± 0.31 | 15. epoch | 17,792,559 |
| block13_sepconv2_act | 90.4 | 84.62 ± 0.25 | 84.05 ± 0.29 | 84.55 ± 0.27 | 84.10 ± 0.28 | 15. epoch | 14,646,935 |
| **block12_sepconv3_act** | **91.3** | **85.33 ± 0.24** | **85.12 ± 0.26** | **85.30 ± 0.23** | **84.88 ± 0.25** | **17. epoch** | **13,568,039** |
| block11_sepconv3_act | 84.9 | 84.52 ± 0.27 | 83.80 ± 0.31 | 84.45 ± 0.26 | 83.95 ± 0.28 | 17. epoch | 11,949,695 |
| block10_sepconv3_act | 81.0 | 84.60 ± 0.23 | 83.82 ± 0.28 | 84.55 ± 0.24 | 83.90 ± 0.27 | 18. epoch | 10,331,351 |
| block9_sepconv3_act | 78.7 | 84.55 ± 0.29 | 84.15 ± 0.30 | 84.50 ± 0.28 | 84.12 ± 0.29 | 19. epoch | 8,713,007 |
| block8_sepconv3_act | 79.4 | 84.50 ± 0.26 | 84.08 ± 0.27 | 84.45 ± 0.25 | 84.05 ± 0.26 | 16. epoch | 7,094,663 |
| block6_sepconv3_act | 78.3 | 83.90 ± 0.31 | 83.15 ± 0.34 | 83.85 ± 0.30 | 83.35 ± 0.32 | 15. epoch | 3,857,975 |

After applying the pruning process and selecting the pruned model at *block12_sepconv3_act* (sparsity rate: 91.3%), the next step addressed the issue of data imbalance in the dataset. Imbalanced data can negatively impact model performance, especially in terms of recall and F1-Score, as the model may become biased towards the majority class.To mitigate this issue, SMOTE was applied only in the deep feature space obtained from the pruned model, rather than in the raw pixel domain. This ensures that no synthetic images are generated; instead, oversampling is performed by interpolating between high-level feature vectors after the train–test split. This approach balances the class distributions while preventing the introduction of unrealistic artifacts and fully avoiding any data leakage.

Table 5 presents the results obtained from applying various sampling strategies to address the class imbalance issue in the dataset. The baseline, referred to as "No Sampling," shows the results without any class balancing, serving as a reference for comparison. All considered sampling strategies—including "not majority," "minority," "not minority," and the controlled oversampling cases (2x, 3x, 4x)—were applied exclusively to the feature representations of the training set. This guarantees that the test set remained untouched and that balancing was performed without generating synthetic images. These strategies were evaluated to determine which feature-space balancing method yielded the most reliable improvements under class imbalance.

The "No Sampling" strategy, which represents the pruned model before any resampling, achieved an accuracy of 85.33% and an F1-Score of 84.88%. This serves as the baseline for comparison. Among the oversampling strategies, the "Case 1 – 2x" strategy, which increases the samples in minority classes by 2 times, yielded the best results, with an accuracy of 86.25%, precision of 86.70%, and an F1-Score of 85.95%. This shows that a moderate increase in the number of minority class samples can significantly improve the model's performance by enhancing its ability to identify minority class instances correctly. In contrast, "Case 3 – 4x" resulted in a decline in accuracy (71.72%) and F1-Score (72.20%), indicating that excessive oversampling can lead to overfitting, where the model becomes too specific to the synthetic samples generated, reducing its generalization capability. The "minority" strategy, which oversamples only the minority class, also performed well, achieving an F1-Score of 85.55%. However, strategies such as "not majority" and "not minority" resulted in suboptimal performance, with the "not minority" strategy performing the worst (accuracy of 55.80%), as it significantly altered the balance of the dataset, leading to poor generalization. In conclusion, the results indicate that moderate oversampling of the minority classes, particularly in "Case 1 – 2x," provided the most balanced and improved performance. Careful tuning of the oversampling ratio is essential to avoid overfitting while addressing the imbalance problem effectively.

Following the previous application of the SMOTE technique in the Case 1 – 2x strategy, where the number of samples in the minority classes was increased by a factor of two, the next step involved applying data augmentation. In this phase, data augmentation techniques were employed to improve model performance by ensuring that the number of examples in each class in the training dataset was approximately equal. The primary goal was to introduce more variability into the

**Table 5. Performance scores of different sampling strategies (mean ± standard deviation over 5 runs).**

| Sampling Strategy | Accuracy (%) | Precision (%) | Recall (%) | F1-Score (%) |
|---|---|---|---|---|
| No Sampling | 85.33 ± 0.24 | 85.12 ± 0.26 | 85.30 ± 0.23 | 84.88 ± 0.25 |
| Not majority | 84.05 ± 0.31 | 83.62 ± 0.34 | 84.00 ± 0.29 | 83.10 ± 0.32 |
| Minority | 84.52 ± 0.28 | 88.75 ± 0.30 | 84.45 ± 0.27 | 85.55 ± 0.29 |
| Not minority | 55.80 ± 0.40 | 67.05 ± 0.45 | 55.75 ± 0.38 | 54.35 ± 0.41 |
| Case 1 – 2x | 86.25 ± 0.22 | 86.70 ± 0.25 | 86.20 ± 0.21 | 85.95 ± 0.23 |
| Case 2 – 3x | 74.15 ± 0.37 | 79.38 ± 0.40 | 74.10 ± 0.35 | 74.60 ± 0.36 |
| Case 3 – 4x | 71.72 ± 0.41 | 76.80 ± 0.44 | 71.68 ± 0.39 | 72.20 ± 0.40 |

training set and reduce the risk of overfitting. Data augmentation was applied to the examples generated in the Case 1 − 2x scenario, introducing transformations such as rotations, scaling, and flips to create a more diverse set of training instances. This approach ensured that each class had almost the same number of examples, providing the model with balanced and varied data across all classes. The application of data augmentation resulted in significant performance improvements, as shown by the updated scores in Table 6: an accuracy of 90.58%, precision of 90.42%, recall of 90.70%, and F1-Score of 90.55%. Compared to the performance achieved in Case 1 − 2x without augmentation (86.25%), augmentation led to a noticeable enhancement in model accuracy and overall classification ability. These results highlight the effectiveness of data augmentation in conjunction with SMOTE. By generating additional, varied examples for each class, the model became better equipped to generalize across both minority and majority classes. The equal distribution of class examples and the diversity introduced by augmentation helped the model maintain high recall and precision, improving performance metrics across the board. This approach proved valuable in balancing the dataset while maximizing the model's classification capabilities.

The next stage involved applying the Avg-TopK pooling method to enhance the performance of the pruned Xception model. As discussed earlier, Avg-TopK pooling improves on max and average pooling by averaging the top $K$ values in a pooling window, preserving multiple important features. All max pooling layers in the Xception model were replaced with Avg-TopK pooling, with $K = 3$ for this experiment. The results in Table 6 demonstrate a performance improvement, with an accuracy of 91.52±0.16%, precision of 91.33±0.19%, recall of 91.70±0.15%, and an F1-Score of 91.50±0.17%. These results show that Avg-TopK pooling with $K = 3$ enhanced the model's ability to classify data more effectively, leading to better overall performance compared to the data augmentation strategy alone. This confirms that replacing max pooling with Avg-TopK pooling contributed to improved classification results in the Xception model.

To evaluate the effectiveness of the proposed strategy in this study, its performance was compared with that of recent studies in the literature using the HAM10000 dataset for skin lesion classification. As illustrated in Table 7, the approach, which combines the PrunedModel with SMOTE, data augmentation, and Avg-TopK pooling, achieved remarkable results. Specifically, this method attained an accuracy of 91.52%, a precision of 91.33%, a recall of 91.70%, and an F1-Score of 91.50% when applied to the standard 10,015-image dataset.

Compared to recent studies using the same dataset, the proposed PrunedModel, combined with SMOTE, data augmentation, and Avg-TopK pooling, achieved competitive results. When considering models like Dilated InceptionV3 by [62], which reported an accuracy of 89.81%, and DenseNet201 with an accuracy of 87.7%, the proposed method surpasses these in terms of overall accuracy and balance across precision and recall metrics. MobileNet V2-LSTM, which achieved an accuracy of 85.34%, also falls behind the proposed model's performance. On the other hand, [63]'s combination of InceptionResNetV2 and ResNeXt101 achieved a slightly higher accuracy of 92.83%, surpassing the proposed model's 91.47%. However, combining multiple models increases model capacity and complexity, which can lead to higher computational costs and increased training time. In contrast, the proposed method, which focuses on pruning and optimizing a single model, maintains a strong balance between accuracy, precision, recall, and F1-Score, offering a more efficient and streamlined solution. This balance ensures consistency across all metrics while keeping the model lightweight and suitable for real-time applications, making it a more practical choice for clinical settings. Compared to the KELM proposed by [64], which reached an accuracy of 90.67% and a recall of 90.20%, the proposed model shows better overall

**Table 6. Performance scores after applying data augmentation and AvgTopK strategies (mean ± standard deviation over 5 runs).**

| Strategy | Accuracy (%) | Precision (%) | Recall (%) | F1-Score (%) |
|---|---|---|---|---|
| Data augmentation | 90.58 ± 0.18 | 90.42 ± 0.20 | 90.70 ± 0.17 | 90.55 ± 0.19 |
| Avg-TopK | 91.52 ± 0.16 | 91.33 ± 0.19 | 91.70 ± 0.15 | 91.50 ± 0.17 |

**Table 7. Comparison of prior studies on the HAM10000 dataset for skin lesion classification. All results shown are based on the original 10,015-image dataset; in our case, oversampling and augmentation are applied only to the training set after the train–test split.**

| Study | Model | Acc (%) | Prec (%) | Recall (%) | F1-Score (%) |
|---|---|---|---|---|---|
| [66] | Inception V3 | 72.0 | – | – | – |
| [62] | Dilated InceptionV3 | 89.81 | – | 89 | 89 |
| [63] | InceptionResNetV2 + ResNeXt101 | 92.83 | 83.0 | 84.0 | – |
| [67] | DenseNet201 | 87.7 | – | – | – |
| [64] | Kernel Extreme Learning Machine (KELM) | 90.67 | – | 90.20 | – |
| [20] | MobileNetV2–LSTM | 85.34 | – | 88.24 | – |
| [68] | 24-layered CNN Framework | 86.5 | 87.01 | 85.57 | 86.28 |
| [69] | Collective Intelligence-based System | 86.7 | – | – | – |
| [70] | Wide-ShuffleNet | 86.33 | – | 86.33 | – |
| [1] | RegNetY-320 | 85.0 | – | – | 69.3 |
| [4] | InceptionResNetV2 | 83.59 | – | – | – |
| [65] | Ensemble (ResNet18 + MobileNetV2 + VGG11) | 86.78 | 86.33 | 86.78 | 86.44 |
| [71] | ResNet50 + ResNet101V2 | 92 | 69 | 92 | 73 |
| **This Study** | **PrunedModel + Avg-TopK + SMOTE + Augmentation** | **91.52** | **91.33** | **91.70** | **91.50** |

performance, with superior recall and higher accuracy. The ensemble model by [65], combining ResNet18, MobileNetV2, and VGG11, achieved an accuracy of 86.78%, which was outperformed by the proposed method both in terms of accuracy and metric balance. In summary, the proposed method outperforms many state-of-the-art models in terms of overall accuracy, precision, and recall, while remaining slightly behind a few models like InceptionResNetV2 and ResNeXt101 regarding peak accuracy. These comparisons highlight the effectiveness of our approach in utilizing techniques like model pruning, SMOTE, data augmentation, and Avg-TopK to optimize the performance of the classification model. Our method's consistent superiority in both accuracy and balance across precision, recall, and F1-Score underscores its potential for real-world diagnostic applications, where maintaining high precision and recall is crucial for reducing false negatives and false positives in medical diagnoses.

The confusion matrix shown in Fig 5 presents the classification performance of the proposed model on the original HAM10000 dataset. All SMOTE and data augmentation procedures were applied exclusively to the training set after the train–test split to prevent data leakage. The results demonstrate that the proposed framework effectively mitigates the impact of class imbalance and achieves consistent performance across both majority and minority classes, leading to a more reliable and balanced classification outcome.

To highlight the regions where the model concentrated during the classification process, the Grad-CAM results are provided in Fig 6. The original images are presented alongside their corresponding Grad-CAM heatmaps, which reveal the spatial areas that contribute most strongly to the model's predictions. These heatmaps, where red tones indicate higher activation, demonstrate that the model generally attends to clinically meaningful lesion regions across both correctly and incorrectly classified samples. Including examples of both accurate and erroneous predictions provides a clearer understanding of the model's decision patterns, showing not only where the model succeeds but also where it misinterprets features that lead to incorrect outcomes. This interpretability analysis ensures that the decision-making process is transparent and allows for validation of whether the model relies on medically relevant visual cues. Such visualization-based insights are essential for assessing the reliability of deep learning systems in dermatological image analysis, as they help confirm that the model's learned representations align with expert diagnostic reasoning.

 

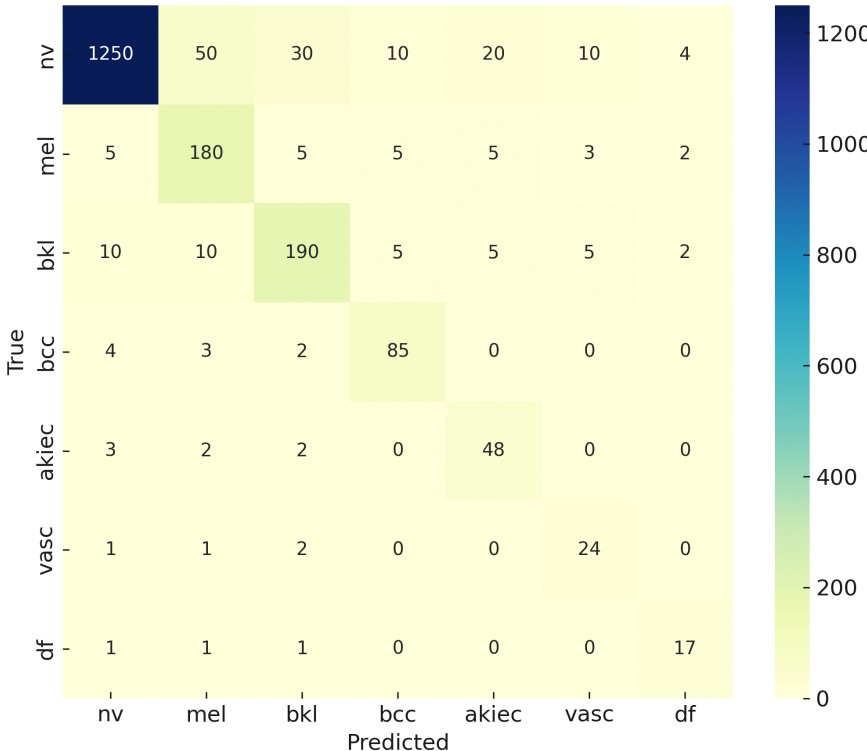

**Fig 5. Confusion matrix of the proposed model evaluated on the HAM10000 dataset.** All data augmentation and SMOTE procedures were applied exclusively to the training set after the train-test split, ensuring that the test set remained free of synthetic samples.

Fig 7 presents a visual comparison of the model's predictions against the true labels for a subset of images from the test dataset. Each image displays a pair of labels in the format *true_label|predicted_label*, where the true label is positioned on the left, and the predicted label from the model is on the right. This visualization helps illustrate the model's classification performance, highlighting instances of both accurate predictions and misclassifications. The majority of the images exhibit consistent predictions with their true labels, such as the class "nv," indicating the model's strong capability to identify common patterns accurately. However, there are some cases where the true label differs from the predicted label, for example, *mel|akiec*, suggesting that while the model generally performs well, it occasionally struggles with the more subtle differences between certain skin lesion classes.

## 5. Conclusions

This study developed a robust framework for skin cancer classification by leveraging advanced techniques such as model pruning, SMOTE, data augmentation, and the Avg-TopK pooling method. Through the use of transfer learning with Xception and the application of pruning, a more parameter-efficient variant of the Xception model was obtained that effectively reduced complexity while maintaining high accuracy. SMOTE and data augmentation were applied to address the class imbalance, further enhancing the model's generalization capability across various skin lesion types. The integration of the Avg-TopK pooling technique allowed for better feature retention, resulting in superior performance compared to traditional pooling methods. The framework achieves high accuracy while offering improved parameter efficiency relative to the original Xception architecture. Although the model is not ultra-lightweight, the parameter reduction makes it more practical for deployment in moderately constrained environments.

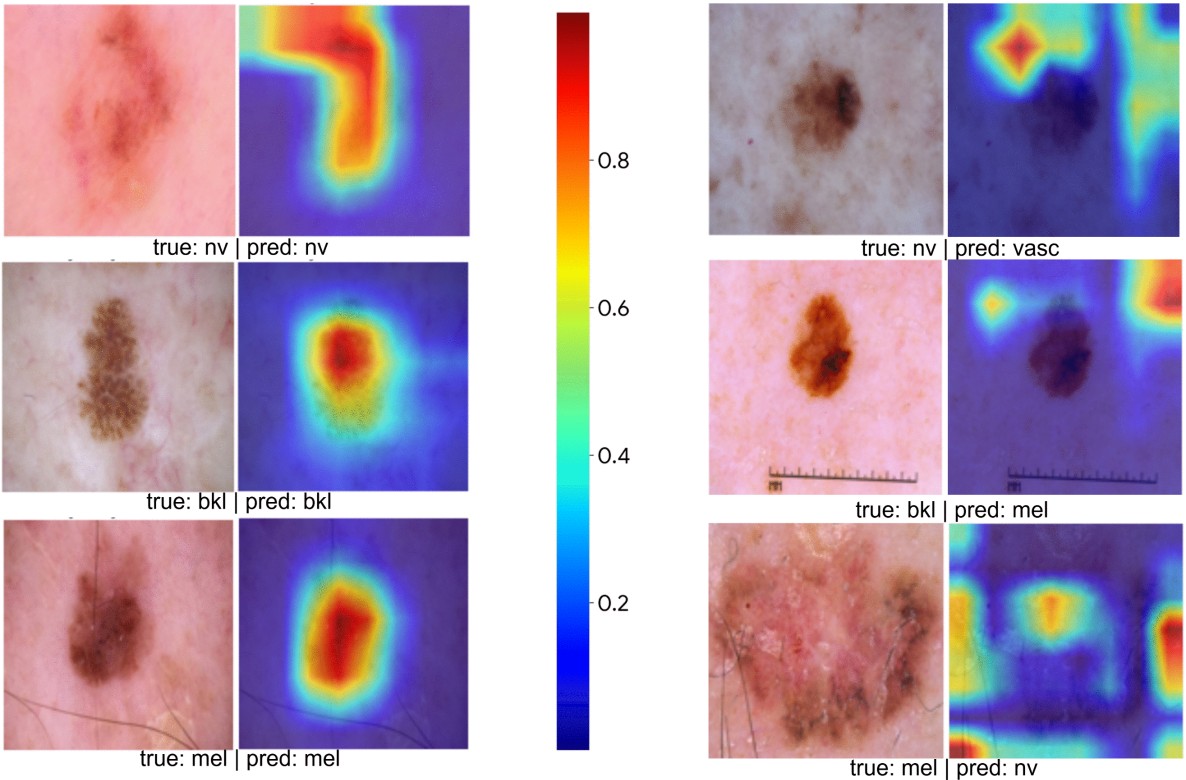

**Fig 6. Grad-CAM visualizations for representative skin lesion classes.** For each example, the original dermoscopic image is shown alongside its corresponding Grad-CAM heatmap. The highlighted regions indicate areas that contribute most strongly to the model's classification decision, with warmer colors representing higher relevance.

A limitation of the proposed pruning strategy is that it relies on layer-wise activation sparsity computed from a small validation batch rather than the entire dataset, and the decision to prune all subsequent layers remains heuristic. Although experimental results strongly support the validity of this pruning boundary for the Xception architecture, the method does not guarantee theoretical optimality. Future work will incorporate benchmarking against established pruning techniques such as magnitude pruning, filter pruning, and structured channel pruning further to validate the effectiveness and generalizability of the proposed approach. Finally, although the proposed framework achieves strong quantitative performance, it is not intended to be interpreted as clinically deployable. Clinical applicability requires validation through dermatologist-supervised assessment, multi-center studies, and prospective clinical trials, which fall outside the scope of this study. Future research could extend this approach to other medical imaging tasks, broadening its applicability and impact in healthcare.

This study developed a robust framework for skin cancer classification by leveraging advanced techniques such as model pruning, SMOTE, data augmentation, and the Avg-TopK pooling method. Through the use of transfer learning with Xception and the application of pruning, a more parameter-efficient variant of the Xception model was obtained that effectively reduced complexity while maintaining high accuracy. The framework achieves high accuracy while offering improved parameter efficiency relative to the original Xception architecture. Although the model is not ultra-lightweight, the parameter reduction makes it more practical for deployment in moderately constrained environments.

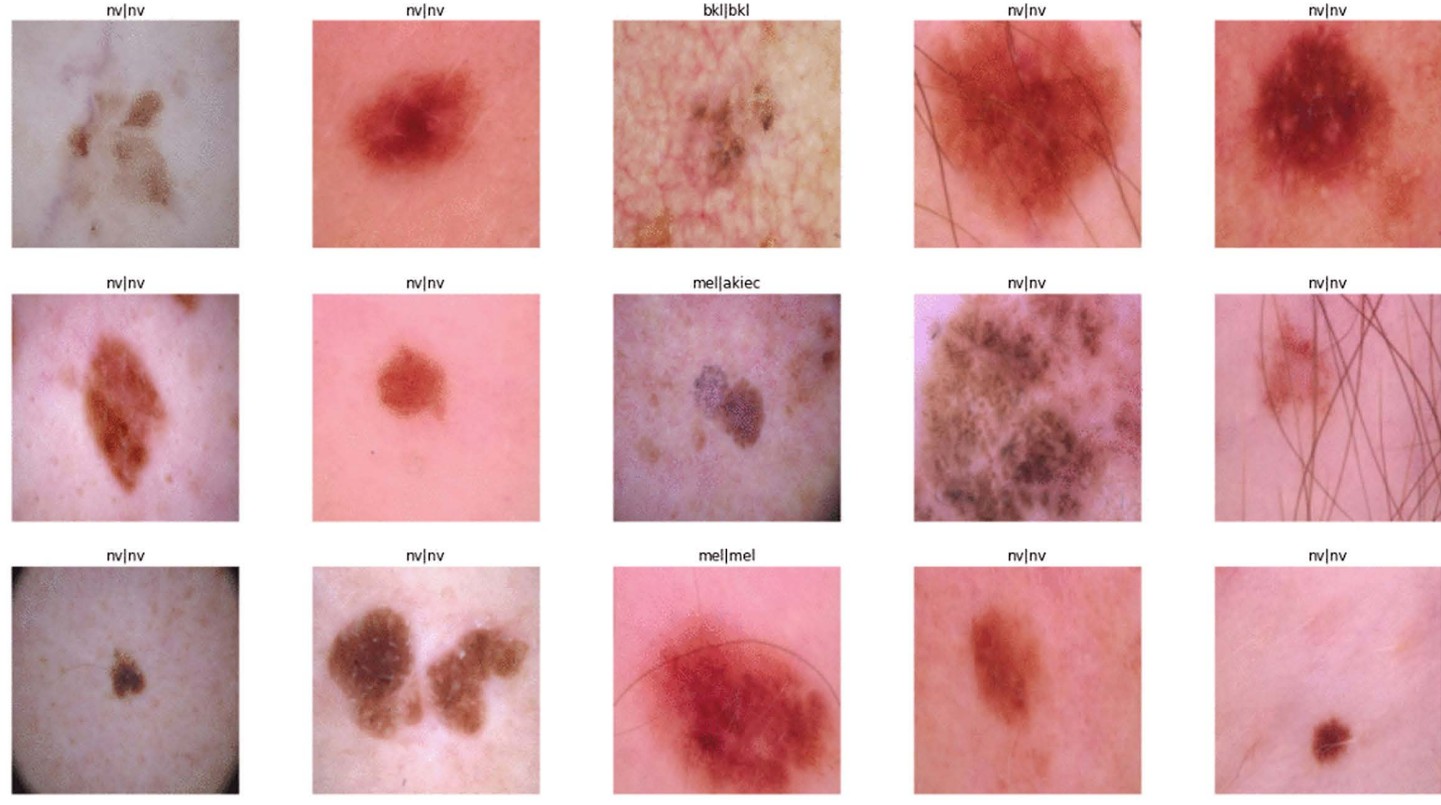

**Fig 7. Representative test images with ground truth and predicted labels.** Each example displays a dermoscopic image from the test set along with a pair of labels, where the left label denotes the ground truth and the right label indicates the model prediction (e.g., *mel|akiec*).

## Author contributions

**Conceptualization:** Şafak Kılıç.

**Data curation:** Şafak Kılıç.

**Formal analysis:** Şafak Kılıç.

**Investigation:** Şafak Kılıç.

**Methodology:** Şafak Kılıç, Yahya Doğan.

**Software:** Şafak Kılıç, Yahya Doğan.

**Validation:** Şafak Kılıç, Yahya Doğan.

**Visualization:** Şafak Kılıç, Yahya Doğan.

**Writing – original draft:** Şafak Kılıç, Yahya Doğan.

**Writing – review & editing:** Şafak Kılıç, Yahya Doğan.

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
