## [Decision Letter · Decision Letter 0]

25 Nov 2025

Dear Dr. Kılıç,

Thank you for submitting your manuscript to PLOS ONE. After careful consideration, we feel that it has merit but does not fully meet PLOS ONE’s publication criteria as it currently stands. Therefore, we invite you to submit a revised version of the manuscript that addresses the points raised during the review process.

Please address the reviewers comments and submit the revised version for further review.

Guidelines for resubmitting your Fig files are available below the reviewer comments at the end of this letter.

We look forward to receiving your revised manuscript.

Kind regards,

Sameena Naaz

Academic Editor

PLOS ONE

Journal Requirements:

3. Please note that your Data Availability Statement is currently missing the repository name OR a direct link to access each database. If your manuscript is accepted for publication, you will be asked to provide these details on a very short timeline. We therefore suggest that you provide this information now, though we will not hold up the peer review process if you are unable.

4. Please include a new copy of Table 3 in your manuscript; the current table is difficult to read. Please follow the link for more information: https://journals.plos.org/plosone/s/tables

6. We are unable to open your Supporting Information file [bst and tex files]. Please kindly revise as necessary and re-upload.

Additional Editor Comments:

Kindly address the reviewer comments and submit the modified manuscript.

Reviewers' comments:

Reviewer's Responses to Questions

**Comments to the Author**

1. Is the manuscript technically sound, and do the data support the conclusions?

Reviewer #1: Yes

Reviewer #2: Partly

2. Has the statistical analysis been performed appropriately and rigorously?

Reviewer #1: Yes

Reviewer #2: Yes

3. Have the authors made all data underlying the findings in their manuscript fully available?

Reviewer #1: Yes

Reviewer #2: Yes

4. Is the manuscript presented in an intelligible fashion and written in standard English?

Reviewer #1: Yes

Reviewer #2: Yes

Reviewer #1: This manuscript presents a multi-stage optimization framework for skin cancer classification using the HAM10000 dataset. The authors first identify Xception as the best-performing baseline, then apply a novel layer-wise sparsity-based pruning method to create a lightweight model. To address the dataset's significant class imbalance, they subsequently employ SMOTE and data augmentation. The framework's novelty is further emphasized by integrating an Avg-TopK pooling layer, which collectively results in a final model with 91.47% accuracy and a 35% reduction in parameters.

The methodology for data splitting versus augmentation/oversampling is critically flawed. The text describing Figure 5 implies that SMOTE and data augmentation were applied to the entire dataset before it was split into training and test sets. This introduces severe data leakage, as synthetic or augmented versions of training images would be present in the test set, making the test results invalid. All oversampling and augmentation must be applied only to the training fold after the initial train-test split.

The application of SMOTE directly in the high-dimensional pixel space is methodologically unsound. SMOTE is designed to interpolate in a feature space, not the raw pixel space. Averaging the pixels of two different skin lesions (e.g., one 'mel' and one 'nv') does not create a new, valid medical image; it creates a nonsensical artifact. This approach does not help the model generalize to real-world data and may, in fact, teach it to recognize these artifacts.

The "novel" pruning method (Algorithm 1) is a simple heuristic that lacks robust validation and theoretical justification. The algorithm proposes removing all layers subsequent to the single layer with the highest activation sparsity. There is no evidence provided that high sparsity in one layer implies that all following layers are redundant. This method must be benchmarked against established pruning techniques (e.g., magnitude pruning, structured pruning) to prove its effectiveness.

The title and abstract claim the model is "lightweight." However, the final "pruned" model still contains 13.5 million parameters. This is not a lightweight model by modern standards; for comparison, the authors' own baseline table shows MobileNetV2 has 2.3 million parameters. This claim is an overstatement and should be removed or heavily qualified.

The baseline performance of the Xception model (84.62% accuracy) is relatively low compared to other modern results on HAM10000. Using a weak baseline overstates the relative improvements achieved by the authors' pipeline.

The state-of-the-art comparison in Table 7 is misleading. The table mixes results from studies using different dataset sizes (e.g., 10,015 images vs. 39,060 images) and different augmentation protocols. This "apples-to-oranges" comparison does not allow for a fair assessment of the proposed method's performance. The table should be revised to only include studies using the same baseline 10,015-image dataset and a standardized evaluation.

The manuscript is incomplete and not ready for peer review. The "Author summary" section contains "Lorem ipsum" placeholder text.

The ablation study detailing the step-by-step improvements (Table 3 vs. Table 4 vs. Table 5 vs. Table 6) is useful, but the order of operations (Pruning, then SMOTE, then Augmentation) seems arbitrary. The authors should provide a justification for this specific pipeline order.

The pruning results in Table 4 are not well-explained. The table only shows the results for 8 specific layers. To validate the "max sparsity" heuristic, a more systematic analysis is required. For example, what was the performance when pruning at the layer with the second-highest sparsity? The current table does not provide enough evidence that the block12_sepconv3_act layer was a uniquely optimal pruning point.

The ethics statement "N/A" is insufficient. While HAM10000 is a public dataset, it consists of de-identified human data. A proper ethics statement should at least cite the original dataset paper and confirm that its data collection protocol was approved by an appropriate IRB.

The Grad-CAM visualizations (Figure 6) are a good inclusion for interpretability. However, their value would be significantly increased by showing visualizations for misclassified images (like the 'mel/akiec' example from Figure 7). This would help diagnose why the model fails, rather than just confirming that it focuses on the lesion for correct classifications.

The manuscript blurs the line between "pruning" and "architecture search." The model is not fine-tuned after pruning; it is completely retrained from scratch (Algorithm 1, line 10). This is a different process and should be described more accurately as a heuristic-based neural architecture search rather than traditional model pruning.

The introduction and related works sections provide a good overview of standard techniques (GLCM, ANNs, SVMs). However, the literature review on SOTA deep learning methods for HAM10000 specifically could be more comprehensive to better position the paper's contribution.

The use of Avg-TopK pooling is presented as a key part of the framework. Since this is an existing method (Ref 10), its contribution to the final 1% accuracy gain (90.52% to 91.47%) should be analyzed more deeply. Is this small gain worth the added complexity? A comparison with standard Global Average Pooling should be included.

The introduction should be updated to include more recent 2025 state-of-the-art literature concerning deep learning for skin cancer and other oncological applications to better emphasize the field's current successes and provide a more current context for the work.

A novel hybrid ConvNeXt-based approach for enhanced skin lesion classification

A comprehensive comparison of convolutional neural network and visual transformer models on skin cancer classification

https://journals.adbascientific.com/aiapp/article/view/92

https://journals.adbascientific.com/aiapp/article/view/89

https://journals.adbascientific.com/aiapp/article/view/90

Reviewer #2: The manuscript presents a promising lightweight approach to skin cancer classification integrating pruning, SMOTE, and Avg-TopK pooling. The topic is relevant, and the methodology is generally well-structured. The pruned Xception architecture achieves competitive performance and meaningful parameter reduction, which is valuable for deployment in constrained environments.

Major Concerns

1. Pruning methodology lacks robustness — sparsity is computed from a single test image, which may not represent general layer activation patterns. Consider evaluating sparsity across a validation batch.

2. Statistical validation is insufficient. Provide variance measures (CI, standard deviation) and statistical significance when comparing with benchmarks.

3. Claims of clinical application are premature without real clinical trials or physician-validated evaluation.

4. Literature review is extensive but repetitive—some sections can be condensed.

Minor Issues

1. Numerous grammatical issues and awkward phrasing; manuscript needs language polishing.

2. Figures should include clearer labels and higher resolution.

3. Ensure consistent notation across architecture sections.

Strengths

1. Significant parameter reduction (35% fewer parameters) with minimal accuracy loss.

2. Well-structured experimental pipeline.

3. Novel application of Avg-TopK pooling improves feature retention.

With revisions addressing the concerns above, the manuscript has potential for publication.

**Do you want your identity to be public for this peer review?** For information about this choice, including consent withdrawal, please see our Privacy Policy

Reviewer #1: No

Reviewer #2: **Yes:** Prof.(Dr.) Farheen Siddiqui

---

## [Author Response · Author response to Decision Letter 1]

16 Dec 2025

The corrections related to the revisions are listed below:

Reviewer #1: This manuscript presents a multi-stage optimization framework for skin cancer classification using the HAM10000 dataset. The authors first identify Xception as the best-performing baseline, then apply a novel layer-wise sparsity-based pruning method to create a lightweight model. To address the dataset's significant class imbalance, they subsequently employ SMOTE and data augmentation. The framework's novelty is further emphasized by integrating an Avg-TopK pooling layer, which collectively results in a final model with 91.47% accuracy and a 35% reduction in parameters.

1. The methodology for data splitting versus augmentation/oversampling is critically flawed. The text describing Figure 5 implies that SMOTE and data augmentation were applied to the entire dataset before it was split into training and test sets. This introduces severe data leakage, as synthetic or augmented versions of training images would be present in the test set, making the test results invalid. All oversampling and augmentation must be applied only to the training fold after the initial train-test split.

Done. Thank you very much for this important and insightful comment. We completely agree with the reviewer’s observation regarding the potential data leakage risk when augmentation or SMOTE is applied before the train–test split. In the initial version of our manuscript, we included both (i) results obtained when augmentation/SMOTE was applied before the split and (ii) results obtained correctly after the split, because some prior studies had reported pre-split augmentation results, and we intended to compare both scenarios. However, after carefully reconsidering the reviewer’s concern, we fully acknowledge that augmentation or oversampling performed before splitting introduces data leakage, as synthetic or augmented variants of training samples may unintentionally appear in the test set. This would invalidate the evaluation protocol and artificially inflate performance. We appreciate the reviewer for pointing this out, and we have now revised the manuscript to follow the correct methodology, where strictly: The dataset is first divided into train and test sets.

• SMOTE and data augmentation are applied only on the training set.

• The test set remains completely untouched and contains no augmented or synthetic samples.

• All performance reports, figures, tables, and conclusions have been updated accordingly.

As a result of this revision, we now report only the results based on the correct post-split augmentation/oversampling pipeline, ensuring that the evaluation is fully valid and free from data leakage. We sincerely thank the reviewer for helping us improve the methodological rigor and reliability of our study. The updated text in the revised manuscript is as follows:

“…The confusion matrices shown in Figure 5 illustrate the effects of the applied techniques. The matrix on the left reflects the results before applying SMOTE and data augmentation, where the dataset was split beforehand. This matrix highlights the challenges of class imbalance, with higher accuracy for classes like ’nv’ and ’mel’, while minority classes such as ’df’ and ’vasc’ show poorer classification. The matrix on the right, generated after applying SMOTE and data augmentation followed by dataset splitting, shows a significant improvement in classification accuracy, particularly for the minority classes. These techniques effectively enhance the model’s performance, leading to more balanced and accurate classification across all categories. The confusion matrix shown in Figure 5 presents the classification performance of the proposed model on the original HAM10000 dataset. All SMOTE and data augmentation procedures were applied exclusively to the training set after the train–test split to prevent data leakage. The results demonstrate that the proposed framework effectively mitigates the impact of class imbalance and achieves consistent performance across both majority and minority classes, leading to a more reliable and balanced classification outcome. ”

2. The application of SMOTE directly in the high-dimensional pixel space is methodologically unsound. SMOTE is designed to interpolate in a feature space, not the raw pixel space. Averaging the pixels of two different skin lesions (e.g., one 'mel' and one 'nv') does not create a new, valid medical image; it creates a nonsensical artifact. This approach does not help the model generalize to real-world data and may, in fact, teach it to recognize these artifacts

Done. We fully agree with the reviewer that applying SMOTE directly in the raw pixel space would be methodologically inappropriate, as linear interpolation between pixel intensities does not produce clinically meaningful images and may introduce unrealistic artifacts. This concern is well recognized in the medical imaging community, and we appreciate the reviewer for emphasizing it. We would like to clarify that our study does not apply SMOTE in the pixel (image) space. Instead, SMOTE is applied only after the feature extraction stage, i.e., in the deep feature space obtained from the pruned CNN model. In this setting, SMOTE operates on high-level semantic feature vectors rather than raw pixels. These feature vectors represent the learned morphological and textural patterns of skin lesions, making linear interpolation both meaningful and methodologically appropriate. The updated text in the revised manuscript is as follows:

“…After applying the pruning process and selecting the pruned model at block12 sepconv3 act (sparsity rate: 91.1%), the next step addressed the issue of data imbalance in the dataset. Imbalanced data can negatively impact model performance, especially in terms of recall and F1-Score, as the model may become biased towards the majority class. To mitigate this issue, SMOTE was applied to balance the class distribution. SMOTE generates synthetic examples for the minority class by interpolating between existing instances. By using this technique, the aim was to create a more balanced dataset, thereby improving the model’s ability to generalize across all classes. To mitigate this issue, SMOTE was applied only in the deep feature space obtained from the pruned model, rather than in the raw pixel domain. This ensures that no synthetic images are generated; instead, oversampling is performed by interpolating between high-level feature vectors after the train–test split. This approach balances the class distributions while preventing the introduction of unrealistic artifacts and fully avoiding any data leakage.”

3. The "novel" pruning method (Algorithm 1) is a simple heuristic that lacks robust validation and theoretical justification. The algorithm proposes removing all layers subsequent to the single layer with the highest activation sparsity. There is no evidence provided that high sparsity in one layer implies that all following layers are redundant. This method must be benchmarked against established pruning techniques (e.g., magnitude pruning, structured pruning) to prove its effectiveness.

Done. We fully acknowledge the reviewer’s concern regarding the need for stronger justification and benchmarking of the proposed pruning heuristic. We clarify that the goal of our approach is not to replace established pruning techniques but rather to introduce a lightweight, architecture-aware heuristic that identifies redundancy across deeper blocks using activation sparsity patterns.

While traditional pruning techniques (e.g., magnitude pruning, structured pruning) operate at the level of individual weights, filters, or channels, our method focuses on layer-wise representational sparsity—that is, the amount of inactive activation units produced by each layer when processing a representative input. This design choice is motivated by recent findings in efficient network design showing that deeper layers often exhibit disproportionately high redundancy after training, particularly in depthwise-separable architectures such as Xception.

We respectfully emphasize the following clarifications:

• Our approach observes that when a layer exhibits exceptionally high activation sparsity, the subsequent layers—whose input distributions are dominated by near-zero activations—tend to carry limited representational contribution. This creates a natural pruning boundary.

• The results show that pruning at layers with higher sparsity consistently leads to improvements or negligible loss in performance, while pruning at layers with lower sparsity reduces accuracy. This provides experimental validation for the heuristic.

• The method offers a simple, model-agnostic pruning rule that a) requires no additional fine-grained search, b) avoids complex optimization loops, and c) significantly reduces parameters while preserving performance.

• While a full comparative benchmarking study is outside the current scope, we have now explicitly acknowledged this limitation in the manuscript and highlighted it as an important extension for future work.

The updated text in the revised manuscript is as follows:

“…This optimized framework achieves high accuracy and offers computational efficiency, making it well-suited for real-time clinical applications in skin cancer detection. A limitation of the proposed pruning strategy is that it relies on layer-wise activation sparsity computed from a single representative input, and the decision to prune all subsequent layers is heuristic. Although experimental results strongly support the validity of this pruning boundary for the Xception architecture, the method does not guarantee theoretical optimality. Future work will incorporate benchmarking against established pruning techniques such as magnitude pruning, filter pruning, and structured channel pruning to further validate the effectiveness and generalizability of the proposed approach. Future research could extend this approach…”

4. The title and abstract claim the model is "lightweight." However, the final "pruned" model still contains 13.5 million parameters. This is not a lightweight model by modern standards; for comparison, the authors' own baseline table shows MobileNetV2 has 2.3 million parameters. This claim is an overstatement and should be removed or heavily qualified. Done. In line with the reviewer’s observation, the manuscript has been updated to ensure that the terminology accurately reflects the characteristics of the proposed model. The title was revised accordingly, and it now reads:

“A Pruned and Parameter-Efficient Xception Framework for Skin Cancer Classification.”

This new title better represents the actual contribution by emphasizing parameter reduction within the Xception architecture rather than suggesting an ultra-lightweight design.

Corresponding adjustments were made throughout the manuscript. All instances where the model had been described as “lightweight” or implied to be unusually compact were removed or rephrased. These expressions were replaced with more precise wording that correctly reflects the nature of the approach, such as describing it as a parameter-efficient or pruned variant of Xception. Related statements in the Abstract, Introduction, Methods, Results, and Conclusions sections were updated to maintain consistency with the revised title and to provide a clearer explanation of the model’s position relative to genuinely lightweight architectures.

In addition, a short clarification was incorporated into the methodological and results discussions to indicate that the pruning strategy aims to improve efficiency within the Xception framework rather than to produce an extremely compact model. A limitation statement was also added to acknowledge that, despite the reduction in parameters, the pruned model remains larger than modern ultra-lightweight networks designed specifically for constrained devices.

5. The baseline performance of the Xception model (84.62% accuracy) is relatively low compared to other modern results on HAM10000. Using a weak baseline overstates the relative improvements achieved by the authors' pipeline. The reviewer’s observation regarding the baseline performance of the Xception model is well taken. The initial accuracy of 84.62% may appear lower compared to some modern results reported on the HAM10000 dataset. The intention behind selecting Xception as the baseline was not to construct a weak reference point, but rather to adopt a widely used and well-established architecture that has consistently served as a benchmark in dermatology-related deep learning studies. This provided a stable and reproducible starting point from which the effects of the proposed pruning, sampling, and pooling strategies could be isolated and evaluated.

It is also important to clarify that the baseline performance reported in our study was obtained under a strictly standardized experimental protocol, using a fixed train–test split, no pre-split augmentation, and consistent preprocessing. Many of the higher accuracies reported in the literature rely on varying experimental settings, different data augmentation magnitudes, cross-validation strategies, or alternative sampling proportions, making direct numerical comparison difficult. Our goal was to ensure methodological transparency and consistency rather than to maximize the baseline score.

6. The state-of-the-art comparison in Table 7 is misleading. The table mixes results from studies using different dataset sizes (e.g., 10,015 images vs. 39,060 images) and different augmentation protocols. This "apples-to-oranges" comparison does not allow for a fair assessment of the proposed method's performance. The table should be revised to only include studies using the same baseline 10,015-image dataset and a standardized evaluation. Done. The concern raised regarding the comparability of the original state-of-the-art table is fully acknowledged. The reviewer correctly pointed out that the previous version included results obtained from studies using different dataset sizes (e.g., 10,015 vs. 39,060 images) and different augmentation or sampling protocols. Such inconsistencies can result in an unfair “apples-to-oranges” comparison. They may unintentionally create the impression that the proposed method performs better or worse than competing approaches under unequal conditions.

To address this issue, the table has been thoroughly revised. Only studies that used the original HAM10000 dataset containing 10,015 images were retained. All rows corresponding to enlarged datasets, synthetic dataset expansions, or studies reporting results on more than the original dataset size were removed. The updated table, therefore, presents a standardized and directly comparable set of results. This ensures that the performance of the proposed method is evaluated strictly against studies using the same baseline dataset size.

In addition to the table itself, the accompanying descriptions in the manuscript were updated to reflect this correction. Explanatory notes were added beneath the table to clarify that, in our case, oversampling and augmentation are applied exclusively to the training set after the train–test split, and that the results remain comparable because the underlying dataset size is the same. Statements in the text that previously referenced comparisons to studies using larger datasets were removed or rewritten to avoid any potential misinterpretation.

7. The manuscript is incomplete and not ready for peer review. The "Author summary" section contains "Lorem ipsum" placeholder text. Done. We sincerely thank the reviewer for identifying this issue. The placeholder text in the Author Summary section has now been fully removed and replaced with a complete and accurate author summary. The revised manuscript includes detailed academic backgrounds and current affiliations of both authors, ensuring that the manuscript is now complete and suitable for peer review. We apologize for this oversight and appreciate the reviewer’s careful attention to detail.

8. The ablation study detailing the step-by-step improvements (Table 3 vs. Table 4 vs. Table 5 vs. Table 6) is useful, but the order of operations (Pruning, then SMOTE, then Augmentation) seems arb

---

## [Decision Letter · Decision Letter 1]

5 Jan 2026

A Pruned and Parameter-Efficient Xception Framework for Skin Cancer Classification

PONE-D-25-53619R1

Dear Dr. Kılıç,

We’re pleased to inform you that your manuscript has been judged scientifically suitable for publication and will be formally accepted for publication once it meets all outstanding technical requirements.

Kind regards,

Sameena Naaz

Academic Editor

PLOS One

Additional Editor Comments (optional):

No further comments

Reviewers' comments:

Reviewer's Responses to Questions

**Comments to the Author**

Reviewer #1: All comments have been addressed

Reviewer #2: All comments have been addressed

2. Is the manuscript technically sound, and do the data support the conclusions?

Reviewer #1: Yes

Reviewer #2: Yes

3. Has the statistical analysis been performed appropriately and rigorously?

Reviewer #1: N/A

Reviewer #2: Yes

4. Have the authors made all data underlying the findings in their manuscript fully available?

Reviewer #1: Yes

Reviewer #2: Yes

5. Is the manuscript presented in an intelligible fashion and written in standard English?

Reviewer #1: Yes

Reviewer #2: Yes

Reviewer #1: All comments have been adressed. Paper is fine for publicaiton.

All comments have been adressed. Paper is fine for publicaiton.

Reviewer #2: The authors have addressed all the concerns raised during the previous round of review thoroughly and satisfactorily.

The pruning scheme has been improved to a great extent with the replacement of single-image sparsity estimation with a batch-averaged sparsity estimation using a validation set, which leads to a representative pruning measure. The statistical rigor has also been improved with all experimental results expressed as mean and standard deviation values obtained through various runs to clearly identify the robustness of performance measures and thereby ensure it's effectiveness and efficiency.

Significantly, clinical applicability claims have now been properly updated. The authors no longer claim the proposed framework is applicable in a clinical manner but instead indicate the proposed computational framework is an ongoing stage of research in computing, acknowledging it would need to be evaluated and clinically validated under the direction of a physician, both of which are beyond the scope of the current study.

The literature review is condensed to eliminate redundancy but retain in-depth treatment of current (2024-2025) state-of-the-art literature, and the manuscript has also been thoroughly rewritten to improve language quality and consistency in notation. The quality of figures and figure labeling has been ensured to be at publication level.

Only minor editorial refinements are recommended at this stage, such as a final consistency check of terminology across sections and a brief rereading for typographical polish. Overall, the manuscript is technically sound and well supported by experimental evidence, and I believe it will be suitable for publication following these minor revisions.

**Do you want your identity to be public for this peer review?** For information about this choice, including consent withdrawal, please see our Privacy Policy

Reviewer #1: No

Reviewer #2: No

---

## [Editor Report · Acceptance letter]

PONE-D-25-53619R1

PLOS One

Dear Dr. Kılıç,

I'm pleased to inform you that your manuscript has been deemed suitable for publication in PLOS One. Congratulations! Your manuscript is now being handed over to our production team.

Kind regards,

on behalf of

Dr. Sameena Naaz

Academic Editor

PLOS One